# Effect of Heat Input on Porosity Defects in a Fiber Laser Welded Socket-Joint Made of Powder Metallurgy Molybdenum Alloy

**DOI:** 10.3390/ma12091433

**Published:** 2019-05-02

**Authors:** Miao-Xia Xie, Yan-Xin Li, Xiang-Tao Shang, Xue-Wu Wang, Jun-Yu Pei

**Affiliations:** 1School of Mechanical and Electrical Engineering, Xi’an University of Architecture and Technology, Xi’an 710055, China; Xiemiaoxia@xauat.edu.cn (M.-X.X.); shang_x_t@163.com (X.-T.S.); 2School of Information Science and Engineering, East China University of Science and Technology, Shanghai 200237, China; wangxuew@ecust.edu.cn; 3State Key Laboratory of Mechanical Behavior for Materials, Xi’an JiaoTong University, Xi’an 710049, China

**Keywords:** molybdenum alloy, fiber laser welding, heat input, porosity defects

## Abstract

Porosity defects are still a challenging issue in the fusion welding of molybdenum and its alloys due to the pre-existing interior defects associated with the powder metallurgy process. Fiber laser welding of end plug and cladding tube made of nanostructured high-strength molybdenum (NS-Mo) alloy was performed in this work with an emphasis on the role of welding heat input. The distribution and morphology of porosity defects in the welded joints were examined by computed tomography (CT) and scanning electron microscopy (SEM). Preliminary results showed that laser welding of NS-Mo under low heat input significantly reduced the porosity defects in the fusion zone. The results of computed tomography (CT) showed that when the welding heat input decreased from 3600 J/cm (i.e., 1200 W, 0.2 m/min) to 250 J/cm (i.e., 2500 W, 6 m/min), the porosity ratio of the NS-Mo joints declined from 10.7% to 2.1%. Notable porosity defects under high heat input were related to the instability of the keyhole, expansion and the merging of bubbles in the molten pool, among which the instability of the keyhole played the dominant role. The porous defects at low heat input were generated as bubbles released from the powder metallurgy base metal (BM) did not have enough time to overflow and escape.

## 1. Introduction

Due to its high fusion point, excellent mechanical properties and low neutron absorption cross-section [1,2,3], nanostructured high-strength molybdenum alloy (NS-Mo) has been selected as one of the candidate materials of the next generation Accident Tolerant Fuel (ATF) to avoid accidents such as that occurring at the Fukushima Nuclear Power Station in 2011. NS-Mo alloy is strengthened by doping with nano-sized rare earth oxide and shows an elongation of about 40% and a strength of about 700 MPa during tensile tests at room-temperature [4]. High-quality and reliable welding technology of NS-Mo must be developed before it can be widely applied in various structures. The appearance of a new molybdenum alloy NS-Mo and its wide application prospects have aroused a research upsurge in welding technology of molybdenum alloys [5,6,7,8,9,10].

Welding methods for Mo alloys include brazing [11,12], electric resistance welding (ERW) [13,14], friction stir welding [15], and fusion welding [16,17,18], etc. The welded seam produced using brazing shows poor high-temperature performance, which does not satisfy the service requirements of cladding tubes at high temperature. Farrell et al. [13] found that electrode sticking and molten metal expulsion occurred during ERW of a molybdenum–rhenium alloy due to the high electrical conductivity and good high-temperature strength of Mo, which resulted in poor-quality joints. Friction stir welding cannot be used to produce small thin-walled components. Fusion welding is a flexible and reliable technique for joining metallic materials, while coarse grains, serious softening, embrittlement, and porosity defects would occur if Mo alloys were joined using the fusion welding method. Wang et al. [16] joined titanium–zirconium–molybdenum (TZM) alloy using tungsten inert gas (TIG) arc welding. They concluded that the microstructure of the fusion zone (FZ) had coarse columnar grains, resulting in reduced joint strength which took up to 37% of that of base metal (BM). X-ray radiographic inspection results showed obvious porosity appeared in the FZ. Considering the high melting point of Mo and its significant tendency of grain coarsening, laser welding technology with the advantages of high energy density is possibly a feasible fusion welding method for Mo alloy [19,20,21,22,23,24].

An et al. [25] investigated the fiber laser welding of end plug and cladding tube made of NS-Mo alloy. It was found that the heat input is the key factor influencing the quality of the NS-Mo laser welded joint. In-depth investigations on the role of heat input in oxygen segregation need to be clarified. Especially, the relationship between heat input and porosity defect has been ignored and remains to be identified. It has been frequently reported that a large welding heat input enhances flotation and removal of bubbles in the molten pool and thereby reduces the porosity defects [26,27,28]. However, some studies indicated that the use of a large heat input cannot guarantee a reduction in porosity defects in Mo alloy welds prepared by powder metallurgy. Stutz et al. [29] observed serious porosity defects when using a large heat input, while welding with a low heat input inhibited pores in the FZ when welding a TZM alloy with a 2 mm thickness using electron beam welding (EBW). Mo alloys are generally prepared using powder metallurgy. In the later phase of sintering the powder metallurgy pieces, small residual pores cannot be eliminated (even with long sintering times) [30], which makes the porosity generated mechanisms of the Mo fusion joint more complicated. Hence, the studies on the formation mechanism of weld porosity under different heat inputs and the influence of the distribution of precipitates on the mechanical properties of the joint are important to be investigated.

In this work, fiber laser welding of end plug and cladding tube made of NS-Mo alloys was carried out with an emphasis on the role of welding heat input. The NS-Mo alloy cladding tubes and end plugs for nuclear applications are prepared by powder metallurgy and hot rolling. The morphologies of welded seams, and porosity defects of the joints between NS-Mo alloy cladding tubes and end plugs prepared using laser welding under quite different heat inputs (i.e. 3600 J/cm and 250 J/cm) were investigated and compared.

## 2. Materials and Methods

The experimental material NS-Mo consists of a Mo metal substrate with 0.5 wt% nano-sized La_2_O_3_ particles dispersed in the substrate. Micro pores are an inevitable problem for powder metallurgy material, as shown in Figure 1. The geometries of the NS-Mo cladding tube and end plug are displayed in Figure 2a. Before welding, the cladding tube and end plug made of NS-Mo were polished, washed in acetone, and then assembled. In order to avoid measurement artifacts from the gap between the end of the cladding tube and the shoulder of the end plug, especially in the porosity measurements, laser beam irradiation of the lap zone of the cladding tube and end plug during welding tests was undertaken, as shown in Figure 2b. The welding tests were carried out using an IPG YLS-4000 fiber laser, as shown in Figure 3, with a maximum output power of 4 kW and a focal spot diameter of 0.2 mm. Before welding, the specimen was preheated to about 450 °C using a resistance wire as the heat source. During welding, the laser head was fixed while the specimen was rotated. Argon was used as the shielding gas. Table 1 shows the welding parameters used in this study.

After welding, the porosity defects in the welded seams were detected using a Y.CT Modular-YXLON industrial computed tomography (CT) system. The CT results were analyzed using the Mimics Medical software. Then, metallographic specimens of the welded joints were prepared using a standard metallographic method and etched with an etchant consisting of 10 mL of HF, 10 mL of HNO_3_, and 30 mL of H_2_O. Next, the cross-sectional morphologies of the welded joints were observed using a Nikon Eclipse MA200 optical microscope (Tokyo, Japan). The porosity morphologies were observed by using an FESEM-SU6600 scanning electron microscope (SEM, HITACHI, Tokyo, Japan).

## 3. Results and Discussion

Welding tests were carried out according to the parameters listed in Table 1. It should be noted that the difference in the heat input of the two joints in this study mainly comes from the significant difference in the welding speed of both joints.

### 3.1. Morphologies of Welded Joints

For convenience of discussing, the circumference of the welded seam is demonstrated in Figure 4. Here, 0° denotes the starting point of welding process and the welding direction is indicated by the red curved arrow. The surface morphologies and cross-sectional morphologies of Joints 1 and 2 are shown in Figure 5. From Figure 5, it can be seen that smooth and regular surface morphology was obtained under lower heat input (i.e., 250 J/cm), while spatters were observed when higher heat input (i.e., 3600 J/cm) was used. It can also be found from Figure 5 that the width of the joint is much less than that of Joint 2.

The cross-section areas of FZ in Joints 1 and 2 (Figure 5) were about 1.01 mm^2^ and 1.76 mm^2^, respectively. As the welding speed of Joint 2 was much lower than that of Joint 1, the length of the molten pool of Joint 2 would be much larger than that of Joint 1. Therefore, the volume of the molten pool of Joint 2 would be much larger than that of Joint 1, leading to more liquid metal surrounding the keyhole and poor stability of the keyhole in the laser welding process of Joint 2. Moreover, the shape and the dynamic behavior of the keyhole are highly dependent on the welding speed during laser welding. Under low heat input and high welding speed, the inclination angle of the front keyhole wall increased significantly. In this case, the laser beam reflected more times on the upper part of the keyhole surface and went through a longer path before reaching the lower part of the keyhole, which was favorable for keeping the upper part of the keyhole open and avoiding excessive concentration of laser energy at the bottom of the keyhole, as shown in Figure 6a. In contrast, under high heat input and low welding speed, the front wall of the keyhole was almost vertical (i.e., the inclination angle of the front keyhole wall was negligible). Therefore, the incident beam easily reached the bottom of the keyhole and excess laser energy may have been concentrated at the lower part of the keyhole, which would cause extensive evaporation of the melt nearby, followed by significant expansion of the lower part of the keyhole, further degrading keyhole stability [31,32,33], as shown in Figure 6b. At the same time, metallic vapor ejected upwards violently from the bottom of the keyhole, forming a large amount of spatters, as shown in Figure 5b. Therefore, the variation of heat input significantly influenced the stability of the keyhole and resulted in the remarkable difference in the weld surface morphology of Joints 1 and 2.

### 3.2. Porosity Defects

In Figure 7, Figure 8 and Figure 9, a comparison of the number, the size, and the typical shape of porosities was made between Joint 1 and Joint 2. In Figure 10, Figure 11, Figure 12 and Figure 13, the formation mechanism of the porosity defects in both joints is demonstrated.

Results of the Computed Tomography (CT) test indicated that welding heat input had significant impact on the number and the size of the porosity defects in fusion welding joints of NS-Mo. Figure 7 reveals that higher heat input results in more porosity defects and larger porosity size. It was found that Joint 1 had about 86 pores of which the diameter mainly ranged from 40 μm to 120 μm, while Joint 2 had 113 pores of which the diameter mainly ranged from 100 μm to 300 μm, as shown in Figure 8. In addition, it can be seen from Figure 7b and Figure 8b that some large pores with the diameter ranging from 300 μm to 500 μm were formed in Joint 2. To objectively evaluate the probability of porosity formation in laser welding processes of NS-Mo, the porosity ratio (i.e., WP) describing the porosity defects per unit volume was defined and expressed as follows: (1)$$\mathrm{WP}\;=\;\frac{V_{P}}{V}$$
where V_P_ and V refer to the total volume of pores in a weld seam and the total volume of the corresponding weld seam zone, respectively. Then, it was calculated that WP of Joint 1 and Joint 2 were 2.1% and 10.7%, respectively.

Results of the SEM test also indicated that the welding heat input had significant impact on the shape of the porous defects in the joints of NS-Mo. As shown in Figure 9a,b, the porosity defects in welded. The porosity defects of Joint 1 achieved under a low heat input mainly appeared as spheres with a smooth internal wall, which were mainly related to the micropores inside the BM. When the micropores were released into molten pool, they were presented likely as spherical bubbles and expanded in the overheated molten pool at atmospheric pressure. Finally, many spherical porosities were left in Joint 1. By contrast, a large number of ellipse-shaped porous defects and many irregularly-shaped pores with large size both appeared in Joint 2 obtained under a high heat input (as shown in Figure 9c,d). Generally, it is considered that irregularly-shaped pores are generated due to a poor stability of the keyhole during deep penetration laser welding (DPLW). The type of pores is very likely to occur during partial penetration DPLW while it is less possible to appear in full penetration DPLW owing to the distribution of laser energy on the keyhole wall which was relatively uniform during full penetration DPLW [34,35]. As discussed above in Section 3.1, keyhole collapse easily occurred during the welding process of Joint 2 due to the fact that laser energy might have excessively concentrated at the bottom of the keyhole. Once the keyhole collapsed, it was likely that irregularly-shaped large porous defects were formed due to the cooling rate of the lower part of the molten pool being very high. By counting the number of pores with a large size (i.e., 100–500 μm) in Joint 2, it was found that there were 33 ellipse-shaped pores and 62 irregularly-shaped pores. Joint 1 was welded under a low heat input at a high welding speed. During the welding process of Joint 1, the stability of the keyhole was favorable so irregularly-shaped porous defects with a large size were basically not observed in Joint 1.

### 3.3. Analysis on Porosity Formation

#### 3.3.1. Inherent Characteristic of Powder Metallurgy Materials

Several studies on EBW of powder metallurgy Mo alloy demonstrated that micropores could significantly aggravate the porosity defects of the welded joint [29,36]. High-pressure gas might be retained in the micropores inside materials in the extrusion process of powder metallurgy materials under 180–200 MPa. During fusion welding, the high-pressure residual gases might be released into the molten pool and expand due to the sudden change of pressure. Additionally, the high temperature of the molten pool can cause further expansion of the bubble. From Figure 1, it was supposed that the diameter of micropores in the BM were about 10 μm. Then large-size bubbles with a diameter of about 99 μm would be formed after expansion of the high-pressure micropores according to the Ideal gas state equation [37]. Therefore, controlling the compactness and impurities of the BMs is an important measure to inhibit the occurrence of porous defects during fusion welding of powder metallurgy materials.

#### 3.3.2. Overflowing of Bubbles in the Molten Pool

Analyzing the behavior of bubble overflow from the molten pool exerts significance on developing effective measures of reducing or eliminating porous defects. In practice, among current measures for controlling porous defects in the fusion welding process, the majority of measures are expected to inhibit porous defects by promoting the escape of bubbles from the molten pool. The sizes of the bubbles and the molten pool are two important factors influencing the time required for the overflow of bubbles from the molten pool. In general, the larger the size of bubbles, the larger is the buoyancy of the bubbles in the molten pool, so the speed of flotation of the bubbles increases and therefore the time required for the bubbles to escape from the molten pool declines. The larger the depth of the molten pool, the longer is the time required for bubbles at the bottom of molten pool to escape. A longer length of molten pool is advantageous for the bubbles to escape from the molten pool. Additionally, the bubbles in the interior of the molten pool require a shorter time to float and overflow from the molten pool, while those in the mushy zone around the solid/liquid interface need a longer time to float and escape from the molten pool owing to being hindered [32]. The floating speed of bubbles in the interior of the molten pool can be calculated according to Stocks formula [38];
(2)$$u=\frac{2\cdot \left(\rho_{1}-{\;\rho }_{2}\right)\cdot g\cdot R^{2}}{9\cdot \eta }$$
where u, ρ_1_, ρ_2_, g, R and η refer to the floating speed of bubbles, the density (i.e., 9.33 kg/m^3^) of Mo molten metal, the density of gas in bubbles, gravitational acceleration (i.e., 9.8 m/s^2^), the radius of bubbles, and the viscosity (i.e., 5.0 × 10^−3^ Pa·s) of the molten Mo alloy, respectively [39]. According to Equation (2), the influence of bubble size on the floating speed of the bubble during the laser welding of Mo alloy and that of the welding heat input (i.e., the size of molten pool) on the time needed for bubbles to escape from the molten pool were roughly estimated, as shown in Figure 10a,b, respectively. The solidification time of the molten pool can be approximately calculated by dividing the length of the molten pool by the welding speed. Based on the surface morphology of the weld seam at the end position, it can be speculated that the lengths of the molten pool during the welding of Joint 1 and Joint 2 were about 1.9 mm and 2.9 mm, respectively, as shown in Figure 11. Thus, during the welding of Joint 1 and Joint 2, the molten metal would be completely solidified after about 0.02 s and 0.87 s, respectively.

It can be speculated from Figure 8 that during the welding of Joint 1, the bubbles with a diameter larger than 120 μm were able to float and overflow from the molten pool. The statistical results of porous defects in Joint 1 also revealed that there were extremely few porous defects with a diameter larger than 120 μm in Joint 1 and the diameter of pores mainly ranged from 40 to 120 μm. It is supposed that the size of a bubble is 50 μm during the welding process of Joint 1, and therefore the floating distance of the bubble is 0.24 mm within 0.02 s which is the typical solidification time in the welding process of Joint 1. Thus, a large number of porous defects were distributed at the lower region of the welded seam in Joint 1, which seemed consistent with the result in Figure 7a. For the few porous defects with a size larger than 120 μm in Joint 1, they were probably related to the fact that high-pressure gas was retained in the pores inside materials in the extrusion process of the powder metallurgy materials under 180–200 MPa. The high-pressure residual gases were released to the molten pool and expanded due to a sudden change of pressure. Additionally, the high temperature of the molten pool can cause further expansion of the bubble [40]. The Ideal gas state equation was used to evaluate the pore diameter after expansion [37]:(3)$$\frac{4}{3}\cdot \pi \cdot R_{2}^{3}=\frac{T_{2}}{T_{1}}\cdot \frac{P_{1}}{P_{2}}\cdot \frac{4}{3}\cdot \pi \cdot R_{1}^{3}$$
here, R_1_ and R_2_ denote the radii of bubbles before and after volume expansion, while P_1_ and P_2_ (i.e., 0.1 MPa) represent the pressures before and after the volume expansion of bubbles in the base metal, respectively. Moreover, T_1_ and T_2_ separately refer to ambient temperature (298 K) and the temperature of the molten pool (i.e., T_2_ ≥ 2895 K). It was assumed that the gas pressure and diameter of micropores in the BMs were 10 MPa and 10 μm, respectively. Therefore, according to Equation (3), it can be deduced that a large-size bubble with a diameter of about 99 μm would be formed after expansion of the high-pressure bubbles.

#### 3.3.3. Formation of Pores with Double Curvature Contour

According to the results shown in Figure 10, it can be seen that all bubbles with a diameter larger than 20 μm were able to overflow out of the molten pool before the rear solid/liquid interface of the molten pool was approached during the welding of Joint 2. However, the statistical results of the porous defects showed that the diameters of the majority of ellipse-shaped porous defects in Joint 2 reached 100–300 μm. Through analysis, the reason for this phenomenon probably was that bubbles generated at the bottom of keyhole easily adhered to the vicinity of the solid/liquid interface and the bubbles in the molten pool merged and expanded. It could be further demonstrated by Figure 12 which shows the computational fluid dynamic (CFD) simulation results of the bubble behavior in the molten pool during the laser welding process. It was found from Figure 12 that bubbles forming near the bottom of the keyhole flowed into the rear part of the molten pool when the keyhole showed a poor stability. Owing to the liquid metals in the mushy zone showing a large viscosity, some bubbles had the probability to adhere and be retained in the vicinity of the solid/liquid interface during the moving process (Figure 12a). Moreover, other bubbles passing by the mushy zone were likely to collide and merge with bubbles retained there (Figure 12b,c). Figure 13 displays the typical double-curvature stomatal contour observed in Joint 2. It can be seen from the figure that the porous defects were likely located in the vicinity of the FZ/HAZ interface, with a size larger than 300 μm. In addition, the contour line of the pores obviously can be divided into two parts according to their curvature. The contour line near to the BMs showed a larger curvature while that far from the BM a smaller one. Through analysis, as the bubbles adhered in the mushy collided and merged with other bubbles, the size of the adhered bubble would increase, while the curvature of the contour away from the solid region would decrease. The curvature of the contour near the solid region was difficult to change due to the high cooling rate there, so eventually pores with a double curvature contour were formed in Joint 2, as shown in Figure 13. In addition, the black elongated morphology in the near junction areas in Figure 13 was the gap between the cladding tube and end plug.

## 4. Conclusions

Fiber laser welding tests of the end plug and cladding tube made of powder metallurgy NS-Mo alloy were carried out and the welding heat inputs were taken as 250 J/cm and 3600 J/cm, respectively. The effects of the welding heat input on porosity defects in the fiber laser welding of powder metallurgy NS-Mo alloy was discussed. The main conclusions were as follows:(1)Laser welding of an NS-Mo alloy under a low heat input resulted in less spatters, and a smooth weld surface. In contrast, a large amount of spatter was observed for samples prepared with a high heat input.(2)When the welding heat input decreased from 3600 J/cm to 250 J/cm, the porosity ratio of the NS-Mo joints declined from 10.7% to 2.1%.(3)Notable irregularly-shaped porosities formed under a high welding heat input were mainly related to the instability of the keyhole, while the spherical defects formed at a low welding heat input were due to the fact that bubbles released from the powder metallurgy BM did not have enough time to escape.

As a recommendation, the authors would like to suggest employing low heat input combined with high welding speed in the fiber laser welding of molybdenum alloy, and suggest strict control of compactness and purity of the powder metallurgy base metal to be welded.

## Figures and Tables

**Figure 1 materials-12-01433-f001:**
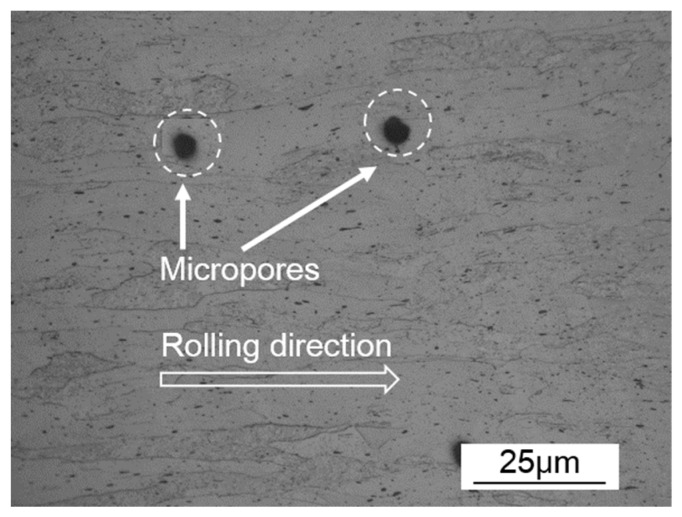
Cross-sectional microstructure of hot-rolled powder metallurgy NS-Mo alloy.

**Figure 2 materials-12-01433-f002:**
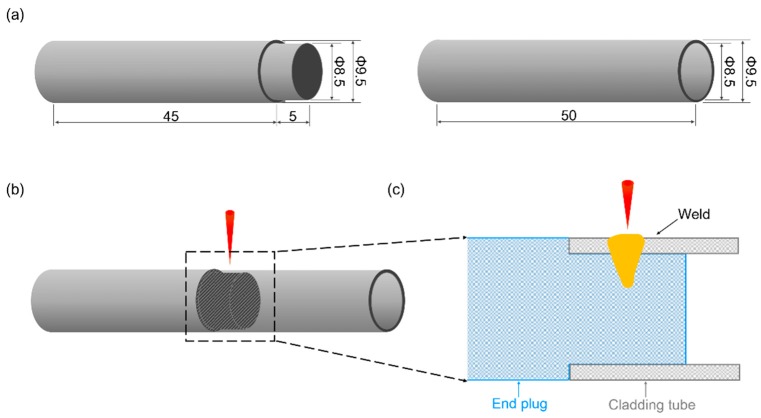
Schematics of: (**a**) dimensions of NS-Mo alloy cladding tube and end plug; (**b**) laser beam irradiation zone on the specimen during welding tests; (**c**) longitudinal section of the sample in the laser irradiation zone. (unit: mm).

**Figure 3 materials-12-01433-f003:**
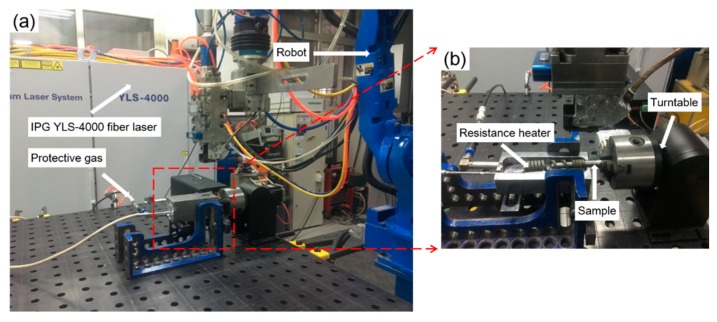
(**a**) Fiber laser welding system; (**b**) tooling and preheating device used in this study.

**Figure 4 materials-12-01433-f004:**
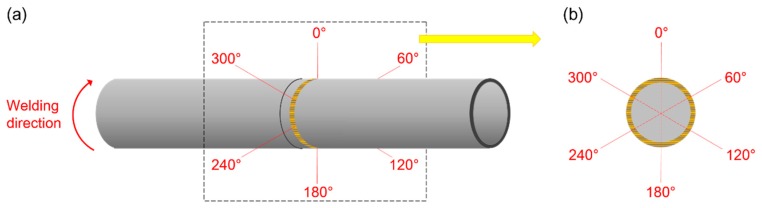
Schematic of the position along the circumference of the tube: (**a**) schematic diagram after welding; (**b**) schematic diagram of angular distribution along the circumference of the tube.

**Figure 5 materials-12-01433-f005:**
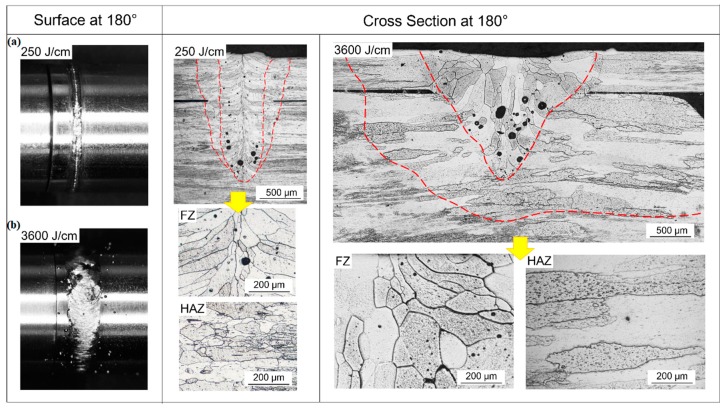
The surface morphology and cross section of NS-Mo alloy joints achieved at the heat input of: (**a**) 250 J/cm; (**b**) 3600 J/cm.

**Figure 6 materials-12-01433-f006:**
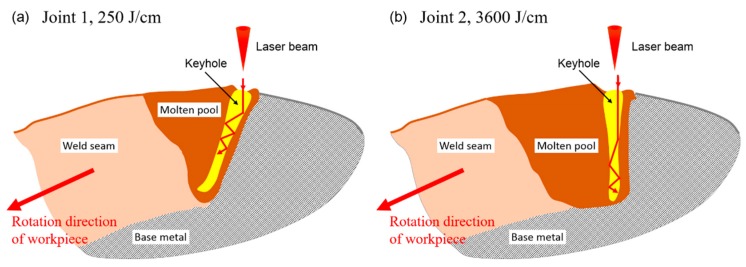
Schematics of laser energy deposition on the keyhole wall during laser welding under: (**a**) low heat input for Joint 1; (**b**) high heat input for Joint 2.

**Figure 7 materials-12-01433-f007:**
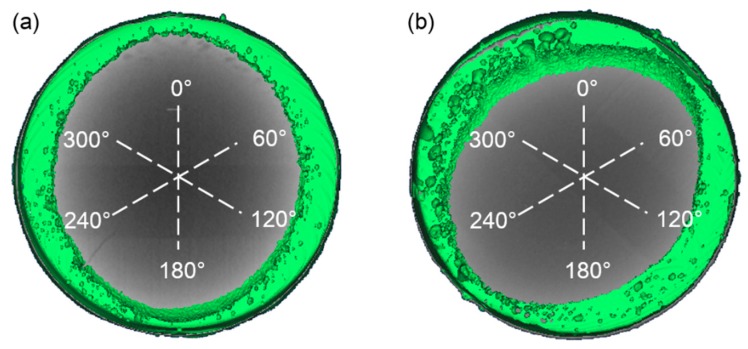
Computed Tomography (CT) test results of porosity defects in NS-Mo alloy joints produced under: (**a**) heat input 250 J/cm; (**b**) heat input 3600 J/cm.

**Figure 8 materials-12-01433-f008:**
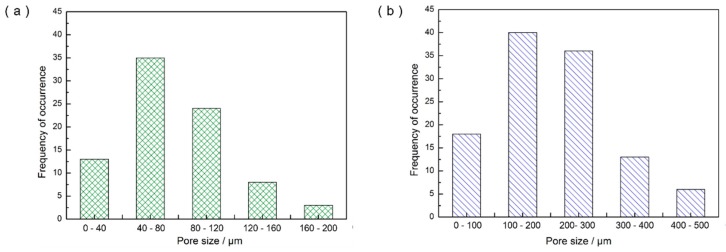
Statistical results of porosity size distribution in laser welding NS-Mo alloy joints produced under: (**a**) heat input 250 J/cm; (**b**) heat input 3600 J/cm.

**Figure 9 materials-12-01433-f009:**
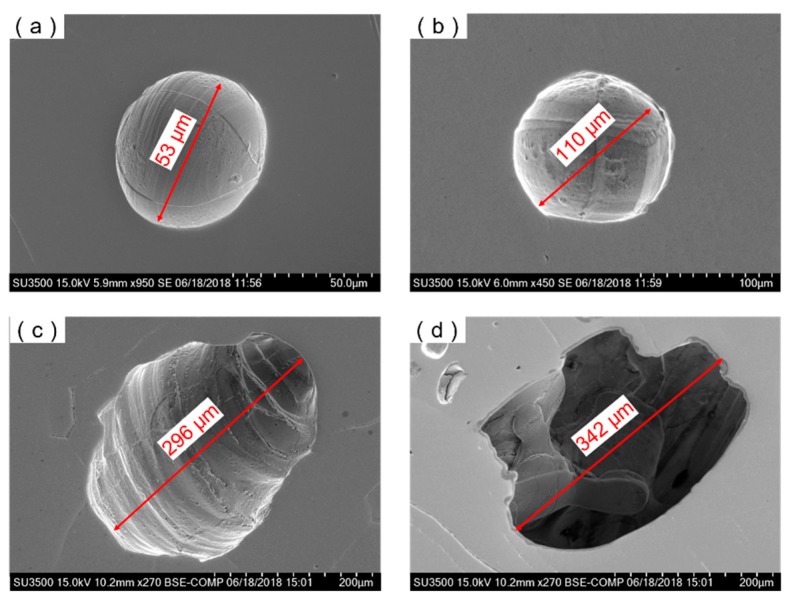
Typical size and shape of pores in laser welding NS-Mo alloy joints produced under: (**a**,**b**) heat input 250 J/cm; (**c**,**d**) heat input 3600 J/cm.

**Figure 10 materials-12-01433-f010:**
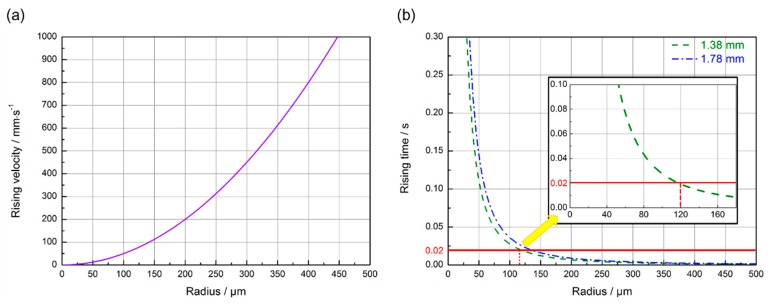
(**a**) The calculated rising velocity of bubbles with different sizes in the NS-Mo alloy molten pool during laser welding; (**b**) escaping time of bubbles with typical size in different molten pool depths.

**Figure 11 materials-12-01433-f011:**
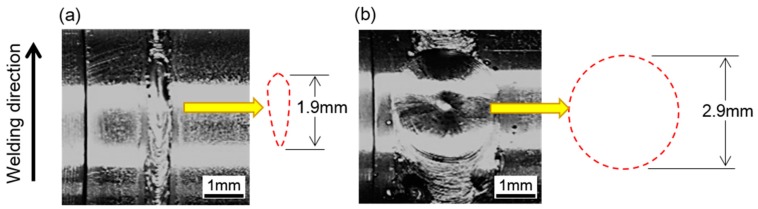
Surface morphology at the end position of: (**a**) Joint 1; (**b**) Joint 2.

**Figure 12 materials-12-01433-f012:**
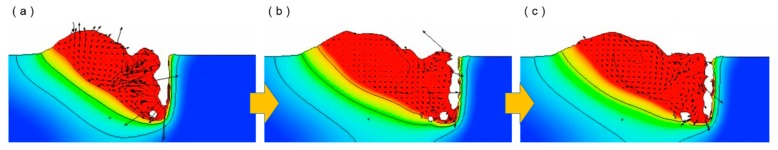
Schematics of bubble merging in the laser welding process with high heat input. (**a**) Bubbles had the probability to adhere and be retained; (**b**,**c**) Bubbles passing by the mushy zone were likely to collide and merge.

**Figure 13 materials-12-01433-f013:**
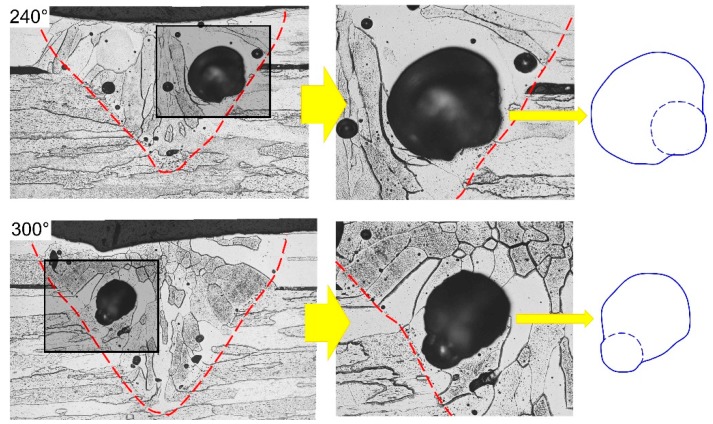
Large size pores with double curvature contour observed in Joint 2.

**Table 1 materials-12-01433-t001:** Laser welding parameters used for NS-Mo alloy.

Sample Number	Heat Input (J/cm)	Laser Power (W)	Welding Speed (m/min)	Defocusing Distance (mm)
Joint 1	250	2500	6	1
Joint 2	3600	1200	0.2	1

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
