# Peer review of "Effect of Heat Input on Porosity Defects in a Fiber Laser Welded Socket-Joint Made of Powder Metallurgy Molybdenum Alloy"

_materials, 2019, doi:10.3390/ma12091433_

Round 1
Reviewer 1 Report
This manuscript presents the results of welding process and studied the welding defects of molybdenum alloy. Study is interesting and I wish it to be published after minor modifications, which should help to improve clearly and quality of presentation.
1) English, especially, in the Abstract and Introduction, should be corrected. For example, multiple verb tense is required in line 3, line 16, and line 21, etc. I recommend one of the professional English editing services.
2) Line 52. What is meaning of An’s study [25]? The first author of the reference [25 ] is Geng. It is better writing Gend [25] found/observed/explained etc… Be sure that the order of Reference list is right.
3) Line 56 “A generally accepted wisdom” seems unscientific.
4) Line 67. Point was missed.
5) Weld junctions are sown, but heat-affected zones did not show in Figure 5 (for example). Please, show heat-affected zones and there structure.
6) Line 120-121.Will measure the junction areas.
7) Explain the microcavities influence to the final defects parameters.
8) Explain , what was a reason of the different shape of the porous in two welding modes.
9) What are the black elongated defects in the near junction areas in Figure 13. Please, clarify.
10) Figure presentations should be improve. Please, make script larger, script color in Fig.7 make white. Clarifications need for Figure 12 and figure 13.
11) Figures captures are not clear; meaning of a, b, etc. should be explained in the captures.
Author Response
Detailed responses to reviewers 1
Title: Effect of heat input on porosity defects in a fiber laser welded socket-joint made of powder metallurgy molybdenum alloy
Response to Reviewer’s comments (highlighted with yellow in manuscript)
Thank you for your time and professional suggestions.
(1)
Comments:
English, especially, in the Abstract and Introduction, should be corrected. For example, multiple verb tense is required in line 3, line 16,and line 21,ect. I recommend one of the professional English editing services.
Response:
Agree.
Please see:
Line 3;
Line 16;
Line 22.
(2)
Comments:
Line 52. What is meaning of An’s study [25]? The first author of the reference[25] is Geng. It is better writing Geng [25] found/observed/explained etc… Be sure that the order of Reference list is right.
Response:
Agree.
Please see:
Lines 54-55.
(3)
Comments:
Line 56 “A generally accepted wisdom” seems unscientific.
Response:
Agree.
Please see:
Lines 58-59.
(4)
Comments:
Line 67. Point was missed.
Response:
Agree.
Please see:
Line 68.
(5)
Comments:
Weld junctions are sown, but heat-affected zones did not show in Figure 5 (for example). Please, show heat-affected zones and there structure.
Response:
Agree.
The Fig.5 has been revised as follow.
Figure 5. The surface morphology and cross section of NS-Mo alloy joints achieved at the heat input of: (a) 250 J/cm; (b) 3600 J/cm.
Please see:
Line 118.
(6)
Comments:
Line 120-121.Will measure the junction areas.
Response:
Agree.
Please see:
Lines 121-125.
(7)
Comments:
Explain the microcavities influence to the final defects parameters.
Response:
Agree.
Please see:
Lines 192-202.
(8)
Comments:
Explain: what was a reason of the different shape of the porous in two welding modes.
Response:
Agree.
Please see:
Lines 171-184.
(9)
Comments:
What are the black elongated defects in the near junction areas in Figure 13. Please, clarify.
Response:
Agree.
The black elongated morphology in the near junction areas in Figure. 13 was the gap between cladding tube and end plug.
Please see:
Lines 279-280.
(10)
Comments:
Figure presentations should be improve. Please, make script larger, script color in Fig.7 make white. Clarifications need for Figure 12 and figure 13.
Response:
Agree.
The Fig.7 has been revised as follow.
Please see:
Line 160;
Lines 263-271;
Lines 274-278.
(11)
Comments:
Figure captions are not clear; meaning of a, b , etc. should be explained in the captures.
Response:
Agree.
The above mentioned content has been modified.
Please see:
Lines 91-93;
Lines 116-117;
Lines 161-162;
Line 165;
Lines 167-168.

Reviewer 2 Report
All my comments are inside the attached paper

Author Response
Detailed responses to reviewers 2
Title: Effect of heat input on porosity defects in a fiber laser welded socket-joint made of powder metallurgy molybdenum alloy
Response to Reviewer’s comments (highlighted with green in manuscript)
Thank you for your suggestions.
(1)
Comments:
As it is known in any article, the author must include here the main techniques of characterization used…, in this study, such as SEM.
Response:
Agree.
Please see
Lines 16-18;
Lines 94-95;
Lines 99-100.
(2)
Comments:
This term is not appropriate?... Which X-RAY? Diffraction ? or
Response:
Agree.
radiographic inspection
Please see:
Line 50.
(3)
Comments:
I think this huge paragraph is not necessary to be in this position of the article. The author must reduce it.
Response:
Agree.
Please see:
Lines 191-201.
(4)
Comments:
The writing style of this reference must be written as other references
Response:
Agree.
Please see:
Line 306.
